# Three-Dimensional Automated, Machine-Learning-Based Left Heart Chamber Metrics: Associations with Prevalent Vascular Risk Factors and Cardiovascular Diseases

**DOI:** 10.3390/jcm11247363

**Published:** 2022-12-12

**Authors:** Andrea Barbieri, Alessandro Albini, Simona Chiusolo, Nicola Forzati, Vera Laus, Anna Maisano, Federico Muto, Matteo Passiatore, Marco Stuani, Laura Torlai Triglia, Marco Vitolo, Valentina Ziveri, Giuseppe Boriani

**Affiliations:** 1Cardiology Division, Department of Biomedical, Metabolic and Neural Sciences, University of Modena and Reggio Emilia, Policlinico di Modena, 41125 Modena, Italy; 22 Clinical and Experimental Medicine Ph.D. Program, University of Modena and Reggio Emilia, 41121 Modena, Italy

**Keywords:** 3D echocardiography, artificial intelligence, cardiac chamber quantification, machine learning

## Abstract

Background. Three-dimensional transthoracic echocardiography (3DE) powered by artificial intelligence provides accurate left chamber quantification in good accordance with cardiac magnetic resonance and has the potential to revolutionize our clinical practice. Aims. To evaluate the association and the independent value of dynamic heart model (DHM)-derived left atrial (LA) and left ventricular (LV) metrics with prevalent vascular risk factors (VRFs) and cardiovascular diseases (CVDs) in a large, unselected population. Materials and Methods. We estimated the association of DHM metrics with VRFs (hypertension, diabetes) and CVDs (atrial fibrillation, stroke, ischemic heart disease, cardiomyopathies, >moderate valvular heart disease/prosthesis), stratified by prevalent disease status: participants without VRFs or CVDs (healthy), with at least one VRFs but without CVDs, and with at least one CVDs. Results. We retrospectively included 1069 subjects (median age 62 [IQR 49–74]; 50.6% women). When comparing VRFs with the healthy, significant difference in maximum and minimum indexed atrial volume (LAVi max and LAVi min), left atrial ejection fraction (LAEF), left ventricular mass/left ventricular end-diastolic volume ratio, and left ventricular global function index (LVGFI) were recorded (*p* < 0.05). In the adjusted logistic regression, LAVi min, LAEF, LV ejection fraction, and LVGFI showed the most robust association (OR 3.03 [95% CI 2.48–3.70], 0.45 [95% CI 0.39–0.51], 0.28 [95% CI 0.22–0.35], and 0.22 [95% CI 0.16–0.28], respectively, with CVDs. Conclusions. The present data suggested that novel 3DE left heart chamber metrics by DHM such as LAEF, LAVi min, and LVGFI can refine our echocardiographic disease discrimination capacity.

## 1. Introduction

Despite the well-known limitations [1], traditional bi-dimensional transthoracic echocardiography (2DE) is still the most often used noninvasive approach in clinical practice for left ventricular (LV) chamber quantification [2], and alternative reference standard methods, specifically cardiac magnetic resonance (CMR), involve high costs and practical challenges for large-scale clinical application.

Yet, many of these issues are now solved by automated three-dimensional echocardiography (3DE) [3]. Recently, a further improvement in 3DE technology provided a new algorithm for left chamber analysis, which is based on the principles of machine learning (dynamic heart model (DHM), Philips Healthcare, Andover, MA, USA) using a training set of over a thousand studies [4], allowing for a feasible, fast, accurate, and reproducible automated quantification of LV and left atrial (LA) size in one single output [5,6,7,8]. Therefore, applying machine learning techniques to 3DE has great potential to revolutionize how we quantify left chambers in our clinical practice. However, while the value of DHM metrics sounds biologically plausible, the multiple automatically measured numerical output indices are not yet corroborated by a demonstrated association with prevalent vascular risk factors (VRFs) and cardiovascular diseases (CVDs).

Accordingly, we sought to evaluate the association of DHM-derived LA and LV metrics with prevalent VRFs and CVDs in a large, unselected population-based cohort and the independent value of LA metrics over LV structure and function measures. Secondarily, to be closer to clinical practice, we reported the values of principal and derived DHM metrics in the healthy population for male and female subgroups, and the intra- and interoperator variability in a random group with heterogeneous acoustic window and case mix.

## 2. Materials and Methods

The study population comprised patients aged >18 years who underwent standard transthoracic Doppler echocardiography for any indication from 14 September 2020 to 9 November 2021 at Modena University Hospital’s echocardiography laboratory. Criteria for enrollment included (1) age > 18 years and (2) complete resting 2D and 3D echocardiographic assessment. We excluded patients with unsatisfactory images—the margins were not seen well and thus deemed untraceable, and patients on dialysis since large fluid shift patients may cause significant extemporaneous differences in LA and LV measurements. Age, sex, height, weight, body surface area (BSA), cardiac rhythm, clinical indications, and history of cardiovascular diseases were recorded at echocardiography. The study protocol followed the ethical guidelines of the 1975 Declaration of Helsinki and was approved by the local ethic committee (Protocol Code: 234-2021, date of approval: 11 May 2021).

### 2.1. Echocardiographic Data

A complete 2D and 3D transthoracic echocardiographic examination was performed, according to current guidelines [9,10], using a commercial ultrasound system (EPIQ CVx, Philips Healthcare) equipped with an X5-1 transducer. We used a single-beat acquisition mode and multiple cardiac cycles (3–5 beats) in patients with atrial fibrillation. Analysis of DHM methodology was described in detail in our recent publication [11].

Briefly, after setting gain, time-gain compensation, and depth on 2D images, a single-beat acquisition mode from the apical 4-chamber view was used to acquire 3D wide-angle datasets. By changing sector width and image depth, the 3D frame rate was optimized. All the acquisitions were performed by operators fully trained in echocardiography with long-standing experience with the 3D technique and trained on echocardiographic datasets focusing on what constitutes adequate automated analysis. The novel vendor software simultaneously detects LV and LA endocardial surfaces using an adaptive analytics algorithm, which uses knowledge-based identification to orient and locate cardiac chambers and patient-specific adaptation of endocardial borders from which LV and LA volumes are derived directly without geometrical assumptions. Using the automated DHM program, which automatically detects LV endo- and epicardial borders at the end-diastole, 3D-LV mass was analyzed, enabling direct LV mass thickness. While it is possible to correct the LV and LA endocardial–epicardial borders at the end-diastole and and-systole, previous studies have shown that LV volumes [4,5,7] and LV mass [12] could be accurately measured using this software, and manual border adjustments led to only clinically insignificant differences.

We considered the following DHM measures: LV end-diastolic volume indexed to BSA (EDVi), LV end-systolic volume indexed to BSA (ESVi), LV ejection fraction (LVEF), LV stroke volume indexed to BSA (SVi), LA maximum volume indexed to BSA (LAVi max), LA minimum volume indexed to BSA (LAVi min), LA ejection fraction (LAEF), automatically calculated as LA maximum volume-LA minimum volume/LA maximum volume), LV mass indexed to BSA, LV mass: LV end-diastolic volume ratio (LVM/LVEDV). We considered the LV global function index (LVGFI) as an additional measure of the LV function originally described with cardiac magnetic resonance [13]. Previous reports have identified LVGFI as a strong predictor of heart failure and CVD events with incremental utility over LVEF [14,15,16]. As per previous descriptions, we defined LVGFI (%) as LV stroke volume/LV global volume × 100, where LV global volume was calculated as the sum of the LV mean cavity volume [(LV end-diastolic volume + LV end-systolic volume)/2] and myocardium volume (LV mass/density). The density of LV was specified as 1.05 g/mL. A higher LVGFI reflects better LV function. 

All 3D echocardiography images were analyzed online using the larger default boundary detection sliders (end-diastolic position = 60/60; end-systolic position = 30/30). This setting defines diastolic and systolic contour positions within the myocardial wall ranging from 0 to 100, 0 being the most inner endocardial contour toward the LV cavity and 100 being the most outer endocardial contour towards the myocardial wall. To the best of our knowledge, there are no specific recommendations about this feature, and the decision is left to the single image laboratory. We deliberately decided to apply these fixed threshold borders settings because they were the ones used in previous validation studies, and they are closer to the settings of CMR [8]. All measurements were entered into an electronic database at the time of the echocardiographic study. No modification from the original database was applied, and no 3D measurement was made offline. Hence, the study consisted of a retrospective analysis of data prospectively entered into the electronic echocardiographic database.

### 2.2. Definition of Vascular Risk Factors and Cardiovascular Disease 

We considered the following VRFs: arterial hypertension, diabetes, and CVDs (prevalent): atrial fibrillation, stroke, ischemic heart disease, cardiomyopathies, >moderate valvular heart disease/prosthesis. To ascertain prevalent VRFs and CVDs, we referred to baseline verbal interviews, patients’ documentation, or records in our institutional database. The diagnosis of these conditions was formulated according to current guidelines.

### 2.3. Reproducibility

To assess the reproducibility of DHM measurements, intraoperator and interoperator variability was evaluated in a subset of 22 randomly selected patients. Two loops from the same patient were acquired and analyzed by a senior physician (A.B.) to evaluate intraoperator variability. Another loop from the same patient was acquired by a junior physician (A.A.) blinded to the previous measurements to assess interoperator variability. All the analyses were performed without contour adjustment (Appendix A).

### 2.4. Statistical Analysis

Data are shown as counts and percentages for categorical variables. Mean ± SD or median and IQR have been used for continuous variables according to the normality or non-normality of the distribution. Comparisons across groups were made using Chi-square or Fisher’s exact test for categorical variables. For continuous variables, parametric or nonparametric tests were utilized according to the normality of the distribution. In particular, independent samples t-test or Mann–Whitney U test was applied for two sample comparisons and analysis of variance or Kruskal–Wallis H test was the choice when more than two groups were analyzed. The independent association of the DHM metrics with VRFs and prevalent CVDs was assessed by multivariable logistic regression models modeling each DHM metric adjusted for confounders (age and sex). 

Intra- and interoperator variability was assessed by performing Pearson’s linear correlation and Bland–Altman analysis (bias and limits of agreement (LOA)) and then reliability was assessed using the interclass correlation coefficient (ICC) and coefficient of variation (CoV).

All tests were two-tailed. Values of *p* < 0.05 were considered statistically significant. All analyses were performed using SPSS version 28.0 for Windows (SPSS, Inc., Chicago, IL, USA). 

## 3. Results

### 3.1. Population Characteristics

A total of 1069 participants for whom DHM data were available were included. The cohort had a median age of 62 [IQR 49–74]; 50.6% were women. The participants with hypertension and diabetes were 37% and 10.9%, respectively. The proportion of participants with prevalent atrial fibrillation, stroke, ischemic heart disease, cardiomyopathies, and >moderate valvular heart disease/prosthesis at the time of DHM was 16%, 4.1%, 14.7%, 11.4%, and 15.7%, respectively (Table 1).

### 3.2. DHM Metrics According to the Presence of VRFs and CVDs

The comparison of DHM metrics among the subsets of participants (i) without VRFs or CVD (healthy), (ii) with VRFs, but without CVD, and (iii) with CVD is displayed in Table 2, Figure 1, and Appendix A. Regarding LV parameters, there was a stepwise decline in LV function and a significant worsening in EDVi, ESVi, and LV mass when considering CVDs compared to the other two groups. Moreover, the VRFs and CVDs subset had higher LVM/LVEDV than the healthy cohort. More concentric LV remodeling patterns were associated with VRFs, while prevalent CVDs were associated with more eccentric LV remodeling, even without reaching statistical significance in the comparison of LVM/LVEDV. No significant difference could be detected comparing SVi among the subgroups. When the SVi was computed with LV global volume to derive LVGFI, this index was significantly different in all the comparisons. When considering the LA parameters (LAVi max, LAVi min, EFLA), significant differences between patients with VRFs and healthy subjects were reported. The CVDs cohort showed the largest LA volumes and the most compromised LA function. 

In the multivariable logistic regression adjusted for age and sex, none of the DHM metrics were significantly associated with prevalent VRFs when assessing healthy and VRFs cohorts. On the contrary, EDVi, ESVi, LVEF, LAEF, LV mass indexed, LAVi max, LAVi min, and LVGFI were independently associated with prevalent CVDs in the adjusted analysis (Figure 2 and Appendix A). LAVi min, LAEF, LVEF and LVGFI showed the most robust association (OR 3.03 [95% CI 2.48–3.70], 0.45 [95% CI 0.39–0.51], 0.28 [95% CI 0.22–0.35], and 0.22 [95% CI 0.16–0.28], respectively, with prevalent CVDs.

### 3.3. Reliability Analysis

Intra- and interoperator variability was assessed in a sample of 22 random patients from the overall population. Of note, this random group had different acoustic window quality (32% suboptimal, 68% optimal) and a heterogeneous case mix, reflecting the complexity of the overall population (32% healthy, 18% VRFs, 50% CVDs), Appendix A. Good linear correlation was assessed by Pearson’s correlation and a low bias with acceptable LOA was reported in the Bland–Altman analysis (Appendix A). No significant trend in the vertical scatter was reported along the values for the principal DHM parameters. ICC and CoV showed excellent intraoperator reliability (Appendix A). When the same analysis was performed by a junior operator, the interoperator variability showed a greater but still acceptable bias with broader LOA. Similarly, reliability analysis showed less agreement even with an ICC greater than 90% for almost all the parameters and acceptable CoV (Appendix A). Notably, atrial morphological and functional parameters showed less consistent intra- and interoperator reliability. 

## 4. Discussion

Our findings add to the literature by demonstrating for the first time the association of DHM metrics with prevalent VRFs and CVDs in a large population of consecutive subjects who underwent echocardiography for any indications. The main findings were: (1) when considering the LA parameters, significant differences between patients with VRFs and healthy subjects were reported; (2) about LV parameters, there was a stepwise decline in LV function and a significant worsening in terms of EDVi, ESVi, and LV mass when considering CVDs; (3) in the multivariable logistic regression analysis, LVEF, LAEF, LAVi min, and LVGFI have the more robust association with prevalent CVDs. Of note, we measured an ICC greater than 90% for most of the parameters and acceptable coefficients of variance with less consistent intra- and interoperator reliability for the LA morphological and functional parameters.

The LA is widely recognized for being extremely sensitive to LV hemodynamic changes [17,18]. Alterations in LA structure and function may occur before observable LV dysfunction, making them potentially more useful for subclinical disease discrimination than LV measures [19]. Indeed, the relationship between 2DE-derived LAVi max and prevalent CVDs has been documented [20]. However, in a large cohort of older community-dwelling adults without prevalent heart failure, ~20% of the population showed LA abnormalities even with normal LAVi max by guidelines [21]. 

One notable result of the present study is that LAVi min and LAEF have the most robust association with prevalent CVDs compared to LAVi max. It may be argued that LAVi max depicts the severity and chronicity of LV filling pressure, which is supported by prognostic data [22]. On the other hand, LAVi max has limitations in detecting early rises in LV filling pressure. It does not predominantly represent LA intrinsic function because it is highly impacted by LV systolic function via mitral plane systolic descent. Conversely, LAVi min, assessed during LV end-diastole (when the LA is immediately exposed to LV end-diastolic pressure), may be more closely associated with LV filling pressure than LAVi max, mirroring a similar relationship between ESVi and LVEF. Consistently, LAVi min is closely associated with LA functional markers, such as LA strains [23], diastolic function [24], and N-terminal pro-B-type natriuretic peptide [25]. Hence, LAVi min has been proposed as a new and perhaps better LA marker, but prognostic data were reported in limited studies [26].

It is important to point out that DHM provides two good LA parameters quickly at the same time (i.e., volume and function), allowing for interesting clinical applications considering that the LA takes time to remodel [27] and may enlarge in a nonspecific way [28]. Indeed, although LA phasic function can be assessed using 2DE volumetric analysis or measuring LA longitudinal strain [17], 3DE eliminates any geometric assumptions regarding LA geometry and is unaffected by the foreshortening of the LA apical view that might occur with 2DE [29]. In our cohort, LAEF was significantly different when comparing VRFs and CVDs to patients without a history of CVDs and VRFs, perhaps reflecting more advanced diastolic dysfunction in these conditions. Our findings strengthen the recent results from the UK Biobank CMR data, which demonstrated the significant associations of LAEF with key prevalent and incident cardiovascular disease outcomes, independently of LV metrics [30]. Moreover, in a study of 169 patients with atrial fibrillation referred for catheter ablation, Inoue et al. similarly demonstrated the association of poorer LAEF by CMR with prior stroke or transient ischemic attack [31]. In addition, previous small studies [32,33] and others using the UK Biobank CMR data [34,35] have demonstrated the association of diabetes with lower LAEF. 

These findings should encourage echocardiographers to incorporate the LAEF evaluation by using 3DE-dedicated software packages. We believe that the feasibility and the clinical implications of the LAVi min and LAEF with DHM support the clinical need to incorporate such novel measures, in addition to the guideline-recommended LA assessment by LAVi max, into a more comprehensive evaluation of LA structure and function. Surprisingly, a recent European Association of Cardiovascular Imaging (EACVI) survey showed that only 10% of centers use 3DE to assess LA volumes and that 68% of centers did not use a measure of LA function [2].

Considering LV parameters, we showed that LVGFI and LVEF provided a more robust association with CVDs, with LVGFI showing the most consistent results. In recent years, the sensitivity and specificity of LVEF for risk stratification have been questioned [36] and novel prognostic indicators, including structural alterations in the LV, have been generated [37]. LVGFI, originally developed utilizing CMR, is an appealing new cardiac parameter supported by proof of a predictive value more significant than EF in different patient populations [13,14,15,38,39,40] integrating global function (LVEF) with heart size, including LV mass. Therefore, the potential clinical use of LVGFI could be maintained with the more precise DHM metrics [5,6,11,41] which are much easier to apply than CMR. However, still being a derived parameter, postprocessing algorithms should hopefully be developed to calculate LVGFI with minimal time delay in terms of transferability to the clinic. 

Given that data for defining reference values for DHM are currently unavailable, we also analyzed the subgroup of healthy subjects in the present study. We found that the mean LV volumes assessed by DHM were larger than contemporary commercial 3DE automated ultrasound imaging systems values [42,43] and more similar to CMR volumes measures [44]. In clinical practice, the LV volume assessment differs greatly depending on whether the operator draws the boundary by the blood tissue interface (i.e., at the tip of the trabeculation) or by the compact myocardium. The latter is closer to the mean value measured by CMR imaging [45]. DHM allows users to choose where the final single endocardial boundary should be, bringing 3DE measurements closer to CMR data. To get this target, we used large default boundary detection sliders (end-diastolic position = 60/60; end-systolic position = 30/30). In this way, DHM minimizes many of the flaws of the traditional 2DE volumetric technique, which is required, for example, for correct quantification of mitral regurgitation and precise diagnosis of the low-flow condition [46]. Table 3 reports the principal DHM metrics for male and female subgroups in the healthy population compared to the normal values of LV and LA size and function according to the recent publications by the World Alliance Societies of Echocardiography Study [42,43] and the reference ranges for cardiovascular magnetic resonance recently published by the Society for Cardiovascular Magnetic Resonance [44]. From this comparison, it is possible to appreciate that the mean LV and LA volumes that we assessed by DHM are bigger than contemporary commercial 3D automated ultrasound imaging systems values and are more comparable to CMR volumes measures when large default boundary detection sliders (end-diastolic position = 60/60; end-systolic position = 30/30) are used. Of note, regarding functional parameters, LVEF and LAEF are comparable across the techniques. Concerning the reference range of indexed LV mass assessed by CMR when papillary muscles are included in LV volume, DHM showed a higher range of normal values. As a result, the mass/volume ratio normal values are higher for DHM echocardiography than for CMR. In contrast, normative values provided for standard 3D echocardiography are lacking. These data suggest, in aggregate, that the reference values for LV chamber volumes assessed with DHM and other contemporary commercial 3D automated ultrasound imaging systems are quite different and cannot be utilized interchangeably. In a short time, scientific societies should be responsible for advising on how to handle the transition from the 2DE to the 3DE era based on the principles of machine learning in the development of echocardiography.

The comparison between different imaging techniques, especially when comparing echocardiography and CMR, requires considering the “imaging quality issue” because 3DE deeply relies on good image acquisition. In our study, we aimed at the highest imaging quality, and the elevated frame rate of the 3D dataset showed that only high-quality data have been analyzed. Our group has already addressed this issue in a recent paper in which 201 consecutive patients underwent DHM echocardiography and 22% were excluded because of an inadequate acoustic window for 3D measurements, and 7% for problematic geometric assumptions [11]. These data are congruent with those published in a recent meta-analysis (pooled prevalence of poor image quality 14.7% (95% CI: 7.1% to 28.2%), combined feasibility 95.8% (95% CI: 88.1 to 98.6%) [47]. It is out of the question, however, that the highest correlation between DHM and CMR is observed in patients with BMI <25 and excellent image quality, even for inexperienced clinicians [48]. Finally, the software is vendor-dependent; thus, results may not be generalized to datasets of other ultrasound vendors.

## 5. Limitations

A more appropriately powered study with the capacity to examine associations would allow for a more granular disease-specific subanalysis.

Due to the observational nature of this study, a significant limitation is the lack of detailed history of other significant factors potentially affecting cardiac remodeling (e.g., hyperlipidemia, obesity, smoking history). This fact may have resulted in the nonsignificant association in the regression analysis. However, diabetes and hypertension are the VRFs most likely to be independently associated with DHM metrics. Indeed, recent data from UK Biobank participants [30] assessed with cardiac magnetic resonance showed that, after fully adjusted logistic regression models considering a greater number of VRFs (hypertension, diabetes, high cholesterol, and smoking), a poorer LA function (lower LAEF) was significantly correlated only with diabetes and smoking, whilst just hypertension was associated with larger LA size. Moreover, hypertension, diabetes, and smoking were associated with significantly poorer LV function by LVGFI. In mutually adjusted models with LVEF instead of LVGFI, diabetes was associated with significantly lower LVEF; associations of LVEF with other VRFs were not statistically significant. 

Moreover, residual confounding or reverse causation cannot be ruled out; nonetheless, the primary goal of this study was to characterize associations rather than make causative inferences.

We do not account for LV and LA global longitudinal strain and N-terminal pro-B-type natriuretic peptide. Strain, however, necessitates additional specialized tracking software since it is not included in the basic DHM on ultrasound machines and reporting systems.

Finally, DHM was not performed on top of conventional 2DE quantitative morphometric evaluation for comparative analysis. However, since the DHM has been at our disposal, 2DE volumetric analysis is no longer routinely performed in our institution.

## 6. Conclusions

The present data suggested that novel 3DE left heart chamber metrics by DHM such as LAEF, LAVi min, and LVGFI can refine our echocardiographic disease discrimination capacity. The following steps will require assessing the associations between DHM metrics with clinical outcomes in additional cohorts and clinical scenarios.

## Figures and Tables

**Figure 1 jcm-11-07363-f001:**
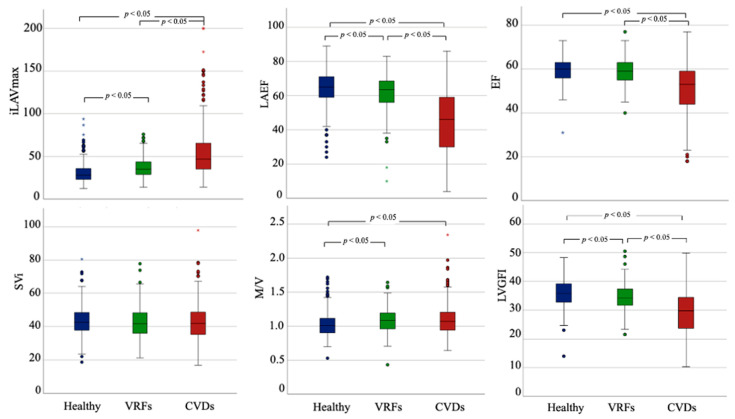
Three-dimensional (3D)-DHM echocardiographic metrics stratified according to the presence of VRFs and CVDs. LAEF = left atrial ejection fraction; LAVi = left atrial volume indexed, LVGFI = left ventricle global function index, M/V = left ventricular mass/left ventricular end-diastolic volume ratio, Svi = stroke volume indexed.

**Figure 2 jcm-11-07363-f002:**
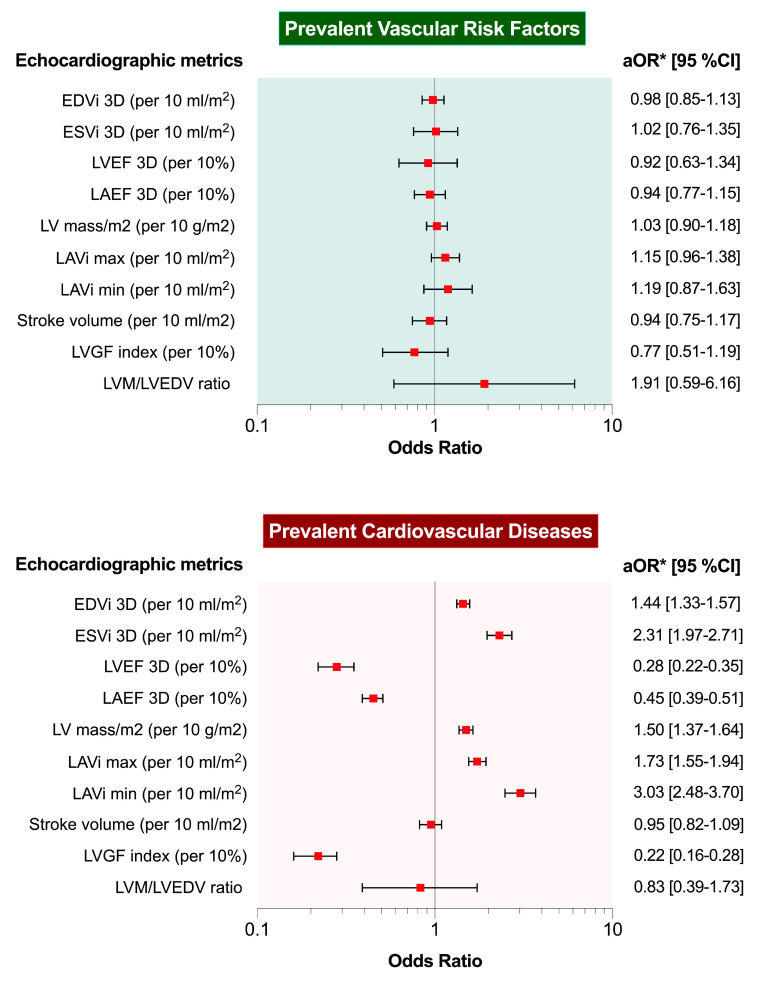
Associations between 3D-DHM echocardiographic metrics and vascular risk factors or prevalent cardiovascular disease. * Adjusted analysis for sex and age; EDVi = end-diastolic volume indexed; ESVi = end-systolic volume indexed; LAEF = left atrial ejection fraction; LAVi = left atrial volume indexed; LVEF = left ventricular ejection fraction, LVGFI = left ventricle global function index, LVM = left ventricular mass, Svi = stroke volume indexed.

**Table 1 jcm-11-07363-t001:** Baseline clinical characteristics stratified by biological sex.

	Men (n = 528; 49.4%)	Women (n = 541; 50.6%)	Total (n = 1069)	*p*-Value
Age (years), median (IQR)	63 (49–74)	61 (49–73)	62 (49–74)	0.18
Weight (Kg), median (IQR)	78 (70–88)	64 (57–72)	70 (62–82)	<0.001
Height (cm), median (IQR)	174 (169–180)	161 (158–165)	167 (150–175)	<0.001
BMI (Kg/m^2^)	25.7 (23.5–28.4)	24.7 (21.6–27.7)	25.2 (22.3–28.1)	<0.001
Hypertension, n (%)	217/528 (41.1)	179/541 (33.1)	396/1069 (37.0)	0.007
Diabetes, n (%)	73/528 (13.8)	43/541 (7.9)	116/1069 (10.9)	0.002
Coronary artery disease, n (%)	121/528 (22.9)	36/541 (6.7)	157/1069 (14.7)	<0.001
DCM, n (%)	48/528 (9.1)	30/541 (5.5)	78/1069 (7.3)	0.02
Acute heart failure, n (%)	28/528 (5.3)	17/541 (3.1)	45/1069 (4.2)	0.08
HCM, n (%)	31/528 (5.9)	13/541 (2.4)	44/1069 (4.1)	0.004
Valvular heart disease, n (%)	86/528 (16.3)	82/541 (15.2)	168/1069 (15.7)	0.61
Moderate/severe MR n (%)	31/528 (6.1)	23/541 (4.3)	55/1069 (5.1)	0.18
Moderate/severe AR n (%)	19/527 (3.6)	12/541 (2.2)	31/1069 (2.9)	0.17
Moderate/severe AS n (%)	12/528 (2.3)	13/539 (2.4)	25/1069 (2.3)	0.88
Moderate/severe TR n (%)	12/528 (2.3)	24/541 (4.4)	36/1069 (3.4)	0.05
Atrial fibrillation, n (%)	106/528 (20.1)	65/541 (12.0)	171/1069 (16.0)	<0.001
Pericarditis/myocarditis, n (%)	10/528 (1.9)	8/541 (1.5)	18/1069 (1.7)	0.59
Previous PE, n (%)	12/528 (2.3)	16/541 (3.0)	28/1069 (2.6)	0.48
COPD, n (%)	21/528 (4.0)	12/541 (2.2)	33/1069 (3.1)	0.09
Previous stroke/TIA, n (%)	27/528 (5.1)	17/541 (3.1)	44/1069 (4.1)	0.11
Previous CHT, n (%)	74/528 (14.0)	162/541 (29.9)	236/1069 (22.1)	<0.001
Malignancy (prior or active), n (%)	138/528 (26.1)	240/541 (44.4)	378/1069 (35.4)	<0.001
Liver disease, n (%)	32/528 (6.1)	12/541 (2.2)	44/1069 (4.1)	0.002

Legend: AR = aortic regurgitation; AS = aortic stenosis; CHT = chemotherapy; CKD = chronic kidney disease; COPD = chronic obstructive pulmonary disease; DCM = dilated cardiomyopathy, including ischemic origin; HCM = hypertrophic cardiomyopathy; IQR = interquartile range.

**Table 2 jcm-11-07363-t002:** Three-dimensional (3D)-DHM echocardiographic metrics stratified according to the presence of VRFs and CVDs.

Echocardiographic Parameters	Total n = 1069	Healthy Subjects (n = 396; 37%)	VRFs Patients (n = 190, 17.8%)	CVDs Patients (n = 483, 45.2%)	*p*-Value *
Frame rate (Hz), median (IQR)	20 (16–22)	21 (20–22)	21 (20–22)	20 (16–21)	<0.001
EDVi 3D, mL/m^2^, median (IQR)	74.6 [63.7–87.6]	71.8 [63.2–81.9]	70.1 [59.8–81.0]	80.6 [66.9–99.5]	<0.001
ESVi 3D, mL/m^2^, median (IQR)	30.8 [25-0-39.5]	28.7 [24.5–33.6]	27.9 [23.1–33.7]	37.1 [27.5–54.2]	<0.001
LVEF 3D (%), median (IQR)	58 [53–62]	60 [56–63]	60 [56–63]	54 [44–59]	<0.001
LAEF 3D (%), median (IQR)	59 [46–67]	65 [59–71]	64 [57–69]	47 [30–59]	<0.001
LV mass/m^2^, median (IQR)	77.9 [67.0–92.2]	70.8 [63.3–81.7]	74.5 [64.1–87.3]	87.7 [74.3–101.8]	<0.001
LAVi, max (mL/m^2^), median (IQR)	35.8 [27.2–49.7]	28.2 [23.2–35.9]	34.9 [29.0–43.4]	46.8 [34.7–65.5]	<0.001
LAVi min (mL/m^2^) median (IQR)	13.9 [9.5–24.8]	9.9 [7.5–13.5]	12.3 [9.8–17.2]	24.4 [14.7–43.7]	<0.001
SVi (mL/m^2^) median (IQR)	42.1 [36.2–48.3]	42.9 [37.9–48.5]	41.4 [35.5–47.4]	42.3 [37.3–48.2]	0.072
LVGFI, median (IQR)	33.6 [29.0–37.4]	35.9 [32.8–39.3]	34.4 [32.0–37.5]	29.7 [23.8–34.6]	<0.001
LVM/LVEDV ratio (g/mL), median (IQR)	1.05 [0.93–1.17]	1.01 [0.90–1.11]	1.08 [0.96–1.19]	1.07 [0.93–1.20]	<0.001

Legend: EDVi = end-diastolic volume indexed; ESVi = end-systolic volume indexed; LAEF = left atrial ejection fraction; LAVi = left atrial volume indexed; LVEF = left ventricular ejection fraction, LVGFI = left ventricle global function index, LVM = left ventricular mass, SVi = stroke volume indexed. * three-ways Kruskal–Wallis test between healthy subjects, VRFs and CVDs.

**Table 3 jcm-11-07363-t003:** Three-dimensional (3D)-DHM echocardiographic metrics in healthy subjects stratified according to sex compared to WASES and SCMR reference values.

Echocardiographic Parameters	3D-DHM (n = 396)		WASES [42,43]		SCMR [44]	
	Men (n = 150, 37.9%)	Women (n = 246, 62.1%)	Men	Women	Men	Women
EDVi, mL/m^2^, m ± SD)	80.4 ± 14.3	68.2 ± 12.0	70 ± 15 §	65 ± 12 §	77 ± 15 *	69 ± 12 *
ESVi, mL/m^2^ m ± SD)	33.6 ± 7.3	26.9 ± 5.9	26 ± 8 §	28 ± 7 §	29 ± 9 *	24 ± 7 *
LVEF (%), m ± SD	58.3 ± 4.3	60.7 ± 4.9	60 ± 5 §	62 ± 5 §	63 ± 6	66 ± 7
LAEF (%), m ± SD	64.4 ± 9.0	64.2 ± 9.1	61.8 ± 7.6 ¶	62.6 ± 7.7 ¶	54 + 8 °	57 + 6 °
LV mass/m^2^, m ± SD	81.1 ± 15.2	69.2 ± 12.9	--	--	56 ± 10 *	45 ± 7 *
LAVi, max (mL/m^2^), m ± SD	33.7 ± 13.8	29 ± 8.8	28.1 ± 7.1 ¶	28 ± 6.7 ¶	41 ± 8 °	44 ± 8 °
LAVi min (mL/m^2^), m ± SD	12.5 ± 7.4	10.5 ± 4.6	10.8 ± 3.7 ¶	10.5 ± 3.7 ¶	19 ± 5 °	19 ± 5 °
SVi (mL/m^2^), m ± SD	47 ± 8.8	41.4 ± 8.3	42 ± 9 §	41 ± 8 §	48 + 9 #	45 + 7 #
LVM/LVEDV ratio (g/mL), m ± SD	1.01 ± 0.16	1.02 ± 0.17	--	--	0.7 ± 0.2 *	0.7 ± 0.1 *

Legend: EDVi = end-diastolic volume indexed; ESVi = end-systolic volume indexed; LAEF = left atrial ejection fraction; LAVi = left atrial volume indexed; LVEF = left ventricular ejection fraction, LVGFI = left ventricle global function index, LVM = left ventricular mass, SVi = stroke volume indexed. WASES: World Alliance Societies of Echocardiography Study; SCMR: Society for Cardiovascular Magnetic Resonance. * papillary muscles included in left ventricular volume (men n = 832, 43%; women n = 1064, 57%) # papillary muscles included in left ventricular volume (men n = 772, 43%; women n = 1004, 57%) ° Simpson’s method; LA appendage excluded (men n = 66, 48%; women = 69, 52%) § according to [42] (men n = 833, 52%; women = 756, 48%) ¶ according to [43] (men n = 901, 51%; women = 864, 49%).

## Data Availability

Data available on request due to restrictions (e.g., privacy or ethical).

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
