# Peer review of "Three-Dimensional Automated, Machine-Learning-Based Left Heart Chamber Metrics: Associations with Prevalent Vascular Risk Factors and Cardiovascular Diseases"

_jcm, 2022, doi:10.3390/jcm11247363_

Round 1

Reviewer 1 Report

The present submission “Three-Dimensional Automated, Machine Learning-Based Left Heart Chamber Metrics: Associations with Prevalent Vascular Risk Factors and Cardiovascular Diseases” is using a large dataset of 3D echocardiographic assessments to evaluate the association and the independent value of DHM-derived LA and LV metrics with prevalent VRFs and CVDs using an automated, machine learning-based Dynamic Heart Model (DHM) to investigate the impact of oxidation on volume overload induced cardiac dysfunction.

Barbieri and colleagues present for the first time the association of DHM metrics with prevalent VRFs and CVDs

Although I’m familiar with cardiac echocardiography including 3D evaluations the technical basics of this submission is beyond my knowledge. Therefore, I’ll evaluate it’s clinical impact.

Major: 

I had difficulties understanding if you were performing TEE or TTE analyses. Please highlight this at various positions in the manuscript including the abstract.

Minor:

Line 16: abbreviation for Dynamic Heart Model is used without explaining the meaning. Please write the whole name instead of abbreviation only

Line 16: same with VRFs, please correct

Line 73: “A complete 2D and 3D echocardiographic examination were performed…” please change to “Complete 2D and 3D echocardiographic examinations were performed

Line 266: please change Alteration to Alterations

Author Response

Reviewer: The present submission “Three-Dimensional Automated, Machine Learning-Based Left Heart Chamber Metrics: Associations with Prevalent Vascular Risk Factors and Cardiovascular Diseases” is using a large dataset of 3D echocardiographic assessments to evaluate the association and the independent value of DHM-derived LA and LV metrics with prevalent VRFs and CVDs using an automated, machine learning-based Dynamic Heart Model (DHM) to investigate the impact of oxidation on volume overload-induced cardiac dysfunction.

Barbieri and colleagues present for the first time the association of DHM metrics with prevalent VRFs and CVDs

Although I’m familiar with cardiac echocardiography including 3D evaluations the technical basics of this submission are beyond my knowledge. Therefore, I’ll evaluate its clinical impact.

Major: 

I had difficulties understanding if you were performing TEE or TTE analyses. Please highlight this at various positions in the manuscript including the abstract.

Authors: Thanks for the suggestion. We specified “transthoracic” throughout the text and the abstract.

Minor:

Line 16: abbreviation for Dynamic Heart Model is used without explaining the meaning. Please write the whole name instead of the abbreviation only

Line 16: same with VRFs, please correct

Line 73: “A complete 2D and 3D echocardiographic examination were performed…” please change to “Complete 2D and 3D echocardiographic examinations were performed

Line 266: please change Alteration to Alterations

Authors: Thanks for the observations. We corrected all the typos and all the abbreviations in the abstract have been explained.

Reviewer 2 Report

This is a well designed study correlating DHM-derived metrics with cardiovascular diseases and vascular risk factors. The text is well written, however there are quite frequent words repetitions, what can be improved. The reliability analysis includes 22 cases. Taking into consideration the complexity of comorbidities, as well as the statistically significant differences in patients' baseline characteristics, it raises the question if 22 cases is enough to assess reliability. Moreover, I suggest adding information about the duration of the presented echocardiography examination, with a comparison to standard 2D echocardiography. 

Minor comments

The abbreviations should be explained when they first appear in text.

The first reference should be before the comma.

References in line 47 should be before the full stop.

References in line 74 should be in one bracket.

References in line 91 should be in one bracket.

Please correct the references in line 103.

The image resolution of figure 2 should be improved.

References in lines: 266, 268, 269 and 271 should appear before full stops and commas.

Please correct the references in lines 298 and 299.

References in line 317 should be in one bracket.

Author Response

Reviewers: This is a well-designed study correlating DHM-derived metrics with cardiovascular diseases and vascular risk factors. The text is well written, however, there are quite frequent word repetitions, which can be improved.

Authors: We reviewed the repeats and changed them following your advice.

Reviewers: The reliability analysis includes 22 cases. Taking into consideration the complexity of comorbidities, as well as the statistically significant differences in patient's baseline characteristics, it raises the question if 22 cases are enough to assess reliability. Moreover, I suggest adding information about the duration of the presented echocardiography examination, with a comparison to standard 2D echocardiography. 

Authors:  thanks to the reviewers for this remark. The sample size was planned regarding the expected width of the 95 percent confidence interval of the ICC. Moreover, in their recent publication on the same topic, Italiano et al reported excellent DHM reliability in a subset of 15 randomly selected patients (DOI: 10.3390/jcm10215030). Such a sample size complies with the indications for optimal designs for reliability studies (DOI: 10.1002/(sici)1097-0258(19980115)17:1<101::aid-sim727>3.0.co;2-e, DOI: 10.1002/sim.935). Moreover, the duration of the echocardiographic examination has been already reported in a recent publication by Italiano et al., which fully corresponds to our daily experience, so we did not perform this evaluation.

Reviewer:

Minor comments

The abbreviations should be explained when they first appear in the text.

Authors: Thanks for the comment. The meaning of each abbreviation used in the text and the abstract have been provided.

Reviewer:

The first reference should be before the comma.

References in line 47 should be before the full stop.

References in line 74 should be in one bracket.

References in line 91 should be in one bracket.

Please correct the references in line 103.

References in lines: 266, 268, 269, and 271 should appear before full stops and commas.

Please correct the references in lines 298 and 299.

References in line 317 should be in one bracket.

Authors: thanks for these observations. All the typos have been corrected.

The image resolution of figure 2 should be improved.

Authors: Thanks for the observation. The image resolution has been improved.

Reviewer 3 Report

TECHNICAL SHORTCOMINGS

 -          In the abstract, in line 16, the abbreviations DHM, LA, LV, VRFs, CVDs are listed without their interpretation.

-           In the text, correct the writing of reference marks; for example: [9], [10] in line 74 should be written [9,10] and [14]- [16]as [14-16], ...

 CONTENT SHORTCOMINGS

 -          In the summary it is stated: VRFs (hypertension, diabetes, elevated cholesterol) ..., and in the methods: arterial hypertension, diabetes, ...

-          Risk factors (diabetes, hypertension, elevated cholesterol) should be more clearly define.

-          Whether all risk factors (e.g. elevated cholesterol and elevated blood pressure) and cardiovascular diseases (e.g. coronary disease and valvular heart disease) have the same effect on the pathogenesis of the examined echocardiographic indicators (e.g. volumes and ejection fractions of LA and LV); whether it is correct to use them as one parameter in data analysis?

-          How important are the drugs that were used in the treatment of the subjects included in the research, or how much drugs such as statins or SGLT2i can affect some of the investigated indicators, e.g. ESVi, LVGFi, ...?

Author Response

Comments and Suggestions for Authors

TECHNICAL SHORTCOMINGS

 -          In the abstract, in line 16, the abbreviations DHM, LA, LV, VRFs, and CVDs are listed without their interpretation.

-           In the text, correct the writing of reference marks; for example: [9], [10] in line 74 should be written [9,10] and [14]- [16]as [14-16], ...

Authors: Thanks for the observations. All the errors were fixed, and the abbreviations used in the abstract have all been defined.

 CONTENT SHORTCOMINGS

 -          In the summary, it is stated: VRFs (hypertension, diabetes, elevated cholesterol) ..., and in the methods: arterial hypertension, diabetes, ...

Authors: Thanks for your observation. We corrected the abstract since the analyzed VRFs were arterial hypertension and diabetes.

-          Risk factors (diabetes, hypertension, elevated cholesterol) should be more clearly defined.

Authors: Thank you for the request. History of arterial hypertension and diabetes mellitus were defined according to the patient’s documentation, interviews, and hospital records. The diagnosis of these conditions was formulated according to current guidelines. We have explained this information in the M&M section.

-          Whether all risk factors (e.g., elevated cholesterol and elevated blood pressure) and cardiovascular diseases (e.g., coronary disease and valvular heart disease) have the same effect on the pathogenesis of the examined echocardiographic indicators (e.g. volumes and ejection fractions of LA and LV); whether it is correct to use them as one parameter in data analysis?

Authors: Thanks for your comment. Recently, data from UK Biobank participants assessed with cardiac magnetic resonance, showed that the cardiovascular risk factors and cardiovascular diseases don’t have the same effect on the examined echocardiographic indicators (Eur Heart J Cardiovasc Imaging 2022 Aug 22;23(9):1191-1200). Furthermore, in clinical practice is important the validation of sensitive echocardiographic parameters to left ventricular hemodynamic changes. making them potentially more useful for subclinical disease discrimination. These are the main reasons why analysis of the association between DHM chamber metrics, vascular risk factors and cardiovascular diseases needs to be analyzed separately

-          How important are the drugs that were used in the treatment of the subjects included in the research, or how much drugs such as statins or SGLT2i can affect some of the investigated indicators, e.g. ESVi, LVGFi, ...?

Authors: Thanks for your thoughtful question. Certainly, medical therapy affects cardiac remodeling in terms of volumetric and functional parameters. However, due to the cross-sectional design of the study, the relative impact of medical therapy was not evaluated and was outside the scope of the study. Therefore, we should assume that medical therapy prescriptions were correct to the requirements and that any incorrect details were well balanced between the groups and attenuated by the sample size.

Reviewer 4 Report

I read with interest "Three-Dimensional Automated, Machine Learning-Based Left Heart Chamber Metrics: Associations with Prevalent Vascular Risk Factors and Cardiovascular Diseases" by Andreas Barbieri et. al.

The authors should be commended for their work on echocardiographic methodology, investigating the Dynamic Heart Model (by Philips) in an unseletcted patient population.

The paper is generally well written, and the data, as well as the statistical analysis seem sound.

However, there are major and minor issues, that I feel need adressing:

Major:

The main issue concerns the study design: 

- Please change all "prospective" aspects into "retrospective". While I second the thought that the echo data was technically recorded prospectively, the entire study was not, since - concerning definition of the groups - you "referred to baseline verbal interviews, patients’ documentation, or record in [your] institutional database.". This is very important, especially with the comments to follow. 

- The comparisons of the echocardiographic parameters is not entirely made clear: what is the point of reference, the "Total", or the "Healthy subjects"? Please indicate this in the text and table 2. (I correctly assume healthy subjects?)

- One main problem is the sub-group of VRF. It is biased, as it lacks many important factors, e.g. smoking, chronic kidney disease, hyperlipoprotenemia, obesity, etc. (compare to e.g. https://doi.org/10.1016/j.mpsur.2018.03.007). Hence, the difference in echo parameters of this sub-group to others (please also see comment above), and the non-significance in the regression model might be viewed as conflicting. So, this must either be well discussed as a whole concept (mentioning "smoking" briefly in the limitations sectionis not sufficient for this problem), or the subgroup must be dropped entirely.

Finally, please add a discussion and / or limitation section on imaging quality. With all justifiable excitement on AI to help us get the "real thing" out of echocardiography, it should not be forgotten that the ultimate prerequisite is to acquire decent images. With low imaging quality, the best AI can only put out low-level results, at best. Most of the time, it will be (dangerous) nonsense. In consequence, in my opinion, a paper on echocardiographic methodology must also always be a paper advocating (a) to strive for the best imaging quality possible, (b) to check the plausibiliy of the results obtained, and (c) to invest in learning / teaching the correct techniques necessary, so (a) and (b) can be achieved. Otherwise we might end-up dropping echocardiography alltogether for a "press-one-button-only"-MRI approach - which is, again in my opinion, already happening, possibly also corroborated by the authors' own statement that "only 10% of centers use 3DE to assess LA volumes and that 68% of centers did not use a measure of LA function.". 

Minor:

- In the Abstract, please define DHM, VRF and CVDs before its first use

- The defined Aim "To evaluate the association and the independent value of DHM-derived LA and LV metrics with prevalent VRFs and CVDs in a large, unselected population of subjects underwent a 3DE automated, machine learning-based Dynamic Heart Model (DHM) using the larger default boundary detection sliders (end-diastolic position = 60/60; end-systolic position = 30/30)" is quite lenghty and hard to read. I suggest to rephrase, maybe split it up.

- Inter- and Intra-observer variability analysis. The authors offer an insight on the imaging in Table S2, concerned with intra- and inter-observer variability. Some points I want to raise: (1) I do not understand the aspect of "intra-oberserver" variability here. Is this not only "inter-observer". (2) we can appreciate that goog care was taken to reach high frame rates in this sample. However, I feel that we need some descriptive statistics on the frame rate in all individuals, since this seems pivotal to correct 3D auto-calculations. (3) You should reference this table already in the corresonding M&M section ("2.3 Reproducibility"). (4) While it is stated in the table's title, I suggest to also define ICC (Interclass Correlation Coefficients) in the legend of Table 3 (it was done for LOA, and , and CoV, as well).

- Please also add info on how many beats were recorded per loop. Especially with 16 % of patients suffering from AFib, a 5-beat acquisition should have been the standard. If not, please discuss. 

- It should be positively highlighted that the authors consequently use indexed measurements for their calculations.

- While we use Philips and the DHM at pur institution, this is not universally applicable. Please explain what "(end-diastolic position = 60/60; end-systolic position = 30/30)" means, especially the numbers. 

- Please rephrase "Data are shown as counts and percentages for categorical variables, mean ± SD or median, and IQR for continuous variables.", by using "parametric" and "non-parametric" data. 

- Please provide BMI and corresponding statistics in Table 1

- Does "Dilated Cardiomyopathy" (DCM) include ischemic origin (sometimes refered to as ICM) or dilation of other causes, only (e.g. myocarditis, genetic disorders, toxitc, etc.)? If DCM includes ICM  (which is correct from a pathophysiological point of view), I suggest to (a) change "Dilated Cardiomyopathy"to "DCM", like done with "HCM" in the same table, and then (b) indicate the context in the legend below, e.g. "DCM= Dilated Cardiomyopathy, including ischemic origin"

- I suggest to move Table 3 to the supplementals, since the study's aim is not primarily to test the DHM performance with different users.  

- The sentence in line 328/329 "DHM allows users to choose where the final single endocardial boundary should be, bringing 3DE measurements closer to CMR data" seems to contradict Lines 89-93: "While it is possible to correct the LV and LA endocardial-epicardial borders at the end-diastole and and-systole, previous studies have shown that LV volumes [4], [5], [7] and LV mass [12] could be accurately measured using this software, and manual border adjustments led to only clinically insignificant differences.". Please elaboarte and/or discuss.

- The sentence from line 347 "In contrast, normative values provided for standard 3D echocardiography are lacking. These data suggest, in aggregate, that the reference values for LV chamber volumes assessed with DHM and other contemporary commercial 3D automated ultrasound imaging systems are quite different and cannot be utilized interchangeably." seems to assume that no data on 3D echocardiographic 3D chamber quantification is present. Please refer to https://doi.org/10.1093/ehjci/jew284, and rephrase.

- Typo in line 73: please use "was" instead of "were performed", since it is about the examination. 

Author Response

I read with interest "Three-Dimensional Automated, Machine Learning-Based Left Heart Chamber Metrics: Associations with Prevalent Vascular Risk Factors and Cardiovascular Diseases" by Andreas Barbieri et. al.

The authors should be commended for their work on echocardiographic methodology, investigating the Dynamic Heart Model (by Philips) in an unselected patient population.

The paper is generally well-written, and the data, as well as the statistical analysis, seem sound.

Authors: We thank the reviewer for the positive comments and interest shown.

However, there are major and minor issues, that I feel need addressing:

Major:

The main issue concerns the study design: 

- Please change all "prospective" aspects into "retrospective". While I second the thought that the echo data was technically recorded prospectively, the entire study was not, since - concerning the definition of the groups - you "referred to baseline verbal interviews, patients’ documentation, or record in [your] institutional database.". This is very important, especially with the comments to follow. 

Authors: Thanks for your thoughtful comment. All measurements were performed online and immediately entered an electronic database at the time of the echocardiogram; no modification from the original database was applied and no measurement was made offline. Hence, the study consisted of a retrospective analysis of data prospectively entered the electronic echocardiographic database. This concept has been better clarified in the methods section.

- The comparisons of the echocardiographic parameters are not entirely made clear: what is the point of reference, the "Total", or the "Healthy subjects"? Please indicate this in the text and table 2. (I correctly assume healthy subjects?)

Authors: Thanks for the observation. The p-value in the table refers to the comparison between the three subgroups, it is the p-value for the three-way comparison according to Kruskall-Wallis analysis. We specified this aspect in the table legend and the methods section has been revised.

- One main problem is the sub-group of VRF. It is biased, as it lacks many important factors, e.g. smoking, chronic kidney disease, hyperlipoproteinemia, obesity, etc. (compare to e.g. https://doi.org/10.1016/j.mpsur.2018.03.007). Hence, the difference in echo parameters of this sub-group to others (please also see comment above), and the non-significance in the regression model might be viewed as conflicting. So, this must either be well discussed as a whole concept (mentioning "smoking" briefly in the limitations section is not sufficient for this problem), or the subgroup must be dropped entirely.

Authors: Thanks for your thoughtful comment. However, diabetes and hypertension diabetes and hypertension are the VRFs most likely to be independently associated with DHM metrics. Indeed, recently data from UK Biobank participants assessed with cardiac magnetic resonance showed that, after fully adjusted logistic regression models considering a greater number of VRFs (hypertension, diabetes, high cholesterol, and smoking), a poorer LA function (lower LAEF) was significantly correlated only with diabetes and smoking, whilst just hypertension was associated with larger LA size. Moreover, hypertension, diabetes, and smoking were associated with significantly poorer LV function by LVGFI. In mutually adjusted models with LVEF instead of LVGFI, diabetes was associated with significantly lower LVEF; associations of LVEF with other VRFs were not statistically significant. (Eur Heart J Cardiovasc Imaging 2022 Aug 22;23(9):1191-1200). We implemented the limitation section of the paper in order to highlight the relevance of lacking some VRFs  in our analysis. 

Finally, please add a discussion and/or limitation section on imaging quality. With all justifiable excitement on AI to help us get the "real thing" out of echocardiography, it should not be forgotten that the ultimate prerequisite is to acquire decent images. With low imaging quality, the best AI can only put out low-level results, at best. Most of the time, it will be (dangerous) nonsense. In consequence, in my opinion, a paper on echocardiographic methodology must also always be a paper advocating (a) to strive for the best imaging quality possible, (b) to check the plausibility of the results obtained, and (c) to invest in learning / teaching the correct techniques necessary, so (a) and (b) can be achieved. Otherwise, we might end up dropping echocardiography altogether for a "press-one-button-only"-MRI approach - which is, again in my opinion, already happening, possibly also corroborated by the authors' statement that "only 10% of centers use 3DE to assess LA volumes and that 68% of centers did not use a measure of LA function.". 

Authors: We thank the reviewer for the stimulating and constructive criticism.  Our group has already addressed many of these correct observations in a recent paper (J. Clin. Med. 2021, 10, 1279. https:// doi.org/10.3390/jcm10061279 ) in which 201 consecutive patients who underwent echocardiography, 22% were excluded because of an inadequate acoustic window for 3D measurements, and 7% for problematic geometric assumptions (feasibility). These data are entirely congruent with those published by a recent meta-analysis (pooled prevalence of poor image quality 14.7 % (95 % CI: 7.1 % to 28.2 %), combined feasibility 95.8 % (95 % CI: 88.1 to 98.6 %) (J Cardiol 2022 Sep 1;S0914-5087(22)00207-6 doi: 10.1016/j.jjcc.2022.08.007). [Online ahead of print]. There is no question, however, that the highest correlation between DHM and CMR is observed in patients with BMI < 25 and excellent image quality, even for unexperienced clinicians (J Echocardiogr 2022 Oct 13. doi: 10.1007/s12574-022-00590-9. [Online ahead of print]. Finally, the software is vendor-dependent; thus, results may not be generalized to datasets of other ultrasound vendors. These concepts have been added in the limitation section.

Minor:

- In the Abstract, please define DHM, VRF, and CVDs before their first use

Authors: Thanks for the observations. All the abbreviations in the abstract have been explained.

- The defined Aim is "To evaluate the association and the independent value of DHM-derived LA and LV metrics with prevalent VRFs and CVDs in a large, unselected population of subjects underwent a 3DE automated, machine learning-based Dynamic Heart Model (DHM) using the larger default boundary detection sliders (end-diastolic position = 60/60; end-systolic position = 30/30)" is quite lengthy and hard to read. I suggest rephrasing, maybe splitting it up.

Authors: Thanks for the observations. We rephrased the sentence as suggested.

- Inter- and Intra-observer variability analysis. The authors offer an insight into the imaging in Table S2, concerned with intra- and inter-observer variability. Some points I want to raise: (1) I do not understand the aspect of "intra-oberserver" variability here. Is this not only "inter-observer". (2) we can appreciate that good care was taken to reach high frame rates in this sample. However, I feel that we need some descriptive statistics on the frame rate in all individuals since this seems pivotal to correct 3D auto-calculations. (3) You should reference this table already in the corresponding M&M section ("2.3 Reproducibility"). (4) While it is stated in the table's title, I suggest also defining ICC (Interclass Correlation Coefficients) in the legend of Table 3 (it was done for LOA, and, and CoV, as well).

Authors: Thanks for your comments. We reported intra-observer variability to highlight that this new technique showed optimal reproducibility when performed by the same operator on the same patient. As suggested, we reported the descriptive statistics with regards of the frame rate in table 2. It should be noted that median frame rate is 20 in the total population, too. The reference to the table was reported in the M&M section and the numeration of Suppl. tables was reset. The legend of table 3 (now supplementary S3, see below) has been corrected as suggested.

- Please also add info on how many beats were recorded per loop. Especially with 16 % of patients suffering from AFib, a 5-beat acquisition should have been the standard. If not, please discuss. 

Authors: Thanks for your question. As stated in the methods section “We used a single-beat acquisition mode and multiple cardiac cycles (3-5 beats) in patients with atrial fibrillation”.

- It should be positively highlighted that the authors consequently use indexed measurements for their calculations.

Authors: We thank the reviewer for these positive comments.

- While we use Philips and the DHM at our institution, this is not universally applicable. Please explain what "(end-diastolic position = 60/60; end-systolic position = 30/30)" means, especially the numbers. 

Authors: Thanks for the comment, which allows us to clarify a fundamental concept. “end-diastolic position = 60/60; end-systolic position = 30/30)" means the left ventricular border setting information. This setting defines diastolic and systolic contour position within the myocardial wall ranging from 0 to 100 being 0 the most inner endocardial contour toward left ventricular cavity and 100 the most outer endocardial contour towards the myocardial wall.  To the best of our knowledge, there are no specific recommendations about this feature, and the decision is left to the single image laboratory. We deliberately decided to apply these fixed threshold borders settings because they were the ones used in previous validation studies and they are closer to the settings of CMR (J. Clin. Med. 2021, 10, 5030. https://doi.org/10.3390/jcm10215030). We have explained these fundamental concepts in the M&M section.

- Please rephrase "Data are shown as counts and percentages for categorical variables, mean ± SD or median, and IQR for continuous variables.", by using "parametric" and "non-parametric" data. 

Authors: Thanks for your comment. The methods section has been revised for clarity reasons. Now, we specified when we applied parametrical and non-parametrical tests according to the normality of the distribution. As reported in the most of research papers, we defined data as categorical and continuous and tests as parametric and non-parametric.

- Please provide BMI and corresponding statistics in Table 1

Authors: Thanks for your suggestion. We added this parameter and its statistics in the table.

- Does "Dilated Cardiomyopathy" (DCM) include ischemic origin (sometimes referred to as ICM) or dilation of other causes, only (e.g. myocarditis, genetic disorders, toxic, etc.)? If DCM includes ICM  (which is correct from a pathophysiological point of view), I suggest (a) changing "Dilated Cardiomyopathy" to "DCM", like done with "HCM" in the same table, and then (b) indicating the context in the legend below, e.g. "DCM= Dilated Cardiomyopathy, including ischemic origin"

Authors: Thanks for the appropriate remark. We confirm that "DCM" also includes ischemic etiology. We have therefore modified the legend of the figure as suggested by the reviewer.

- I suggest moving Table 3 to the supplementals since the study's aim is not primarily to test the DHM performance with different users.  

Authors: thanks for your suggestion. We moved table 3 to the supplementary material (now table S3).

- The sentence in lines 328/329 "DHM allows users to choose where the final single endocardial boundary should be, bringing 3DE measurements closer to CMR data" seems to contradict Lines 89-93: "While it is possible to correct the LV and LA endocardial-epicardial borders at the end-diastole and and-systole, previous studies have shown that LV volumes [4], [5], [7] and LV mass [12] could be accurately measured using this software, and manual border adjustments led to only clinically insignificant differences.". Please elaborate and/or discuss.

Authors: Thanks for the observation, as it gives us a chance to further clarify that, as explained above, the single image laboratory should first decide the fixed threshold borders settings. After that, for each individual case, the operator has the possibility of making small changes to the track if he deems it necessary; these small changes, however, are infrequent in clinical practice and have been shown to not substantially affect the final output.

- The sentence from line 347 "In contrast, normative values provided for standard 3D echocardiography are lacking. These data suggest, in aggregate, that the reference values for LV chamber volumes assessed with DHM and other contemporary commercial 3D automated ultrasound imaging systems are quite different and cannot be utilized interchangeably." this seems to assume that no data on 3D echocardiographic 3D chamber quantification is present. Please refer to https://doi.org/10.1093/ehjci/jew284, and rephrase.

Authors: Thanks for the observation. The first sentence refers to ventricular mass and LV mass/volume ratio whose normal values are not available in the WASES study and the EACVI NORRE study.

- Typo in line 73: please use "was" instead of "were performed", since it is about the examination. 

Authors: Thanks for the observations. We corrected the typo.

Round 2

Reviewer 2 Report

the authors have addressed the comments and improved the manuscript sufficiently

Author Response

We thank the reviewer for his/her comment.

Author Response

We thank the reviewer for his/her comment

Reviewer 4 Report

I have read the revision of "Three-Dimensional Automated, Machine Learning-Based Left Heart Chamber Metrics: Associations with Prevalent Vascular Risk Factors and Cardiovascular Diseases" by Andrea Barbieri et. al.

The authors have adequately addressed all my comments. Thank you.

It seems that some minor typos / spelling errors have been introduced in the Limitatins section:

- "...about others significant factors..." => please change to "...other significant factors..."

- "diabetes and hypertension" seems doubled (Line 475)

- "Indeed, recently data from UK Biobank participants" => please change to "Indeed, recent data from UK Biobank participants"

Please re-check the manuscript for other spelling errors.

Author Response

We thank the reviewers for the useful comments and suggestions. All the changes are reported in the “track changes” version of the manuscript and reported below.

The authors have adequately addressed all my comments. Thank you.

It seems that some minor typos / spelling errors have been introduced in the Limitatins section:

- "...about others significant factors..." => please change to "...other significant factors..."

- "diabetes and hypertension" seems doubled (Line 475)

- "Indeed, recently data from UK Biobank participants" => please change to "Indeed, recent data from UK Biobank participants"

Please re-check the manuscript for other spelling errors.

Authors: thanks for these observations. All the typos have been corrected.